# The Bibliometric Landscape of Gene Editing Innovation and Regulation in the Worldwide

**DOI:** 10.3390/cells11172682

**Published:** 2022-08-29

**Authors:** Xun Wei, Aqing Pu, Qianqian Liu, Quancan Hou, Yong Zhang, Xueli An, Yan Long, Yilin Jiang, Zhenying Dong, Suowei Wu, Xiangyuan Wan

**Affiliations:** 1Zhongzhi International Institute of Agricultural Biosciences, Research Center of Biology and Agriculture, Shunde Graduate School, University of Science and Technology Beijing, Beijing 100024, China; 2Beijing Beike Institute of Precision Medicine and Health Technology, Beijing 100192, China

**Keywords:** gene editing technology, bibliometrics analysis, text clustering, CRISPR, regulation

## Abstract

Gene editing (GE) has become one of the mainstream bioengineering technologies over the past two decades, mainly fueled by the rapid development of the CRISPR/Cas system since 2012. To date, plenty of articles related to the progress and applications of GE have been published globally, but the objective, quantitative and comprehensive investigations of them are relatively few. Here, 13,980 research articles and reviews published since 1999 were collected by using GE-related queries in the Web of Science. We used bibliometric analysis to investigate the competitiveness and cooperation of leading countries, influential affiliations, and prolific authors. Text clustering methods were used to assess technical trends and research hotspots dynamically. The global application status and regulatory framework were also summarized. This analysis illustrates the bottleneck of the GE innovation and provides insights into the future trajectory of development and application of the technology in various fields, which will be helpful for the popularization of gene editing technology.

## 1. Introduction

At present, a new round of global scientific and technological revolution is blooming and reconstructing the global innovation landscape. The rapid development of biotechnology and integration with information technology has become a pillar of driving power. As one of the subversive achievements in the field of life science, gene editing technology is not only a biological research tool but also an important means to improve agricultural production and the healthcare industry. It will play indispensable roles in crop improvement, livestock and poultry breeding, gene-targeted therapy, virus vaccine production and many related fields.

Gene editing technology refers to a rising biological technology that can “Edit” the target genes and achieve the knockout and addition of specific DNA fragments in the genome. It can perform targeted and effective gene modifications in eukaryotes, especially mammals [1]. Gene editing technology is mainly represented by meganuclease, zinc finger nucleases (ZFNs), transcription activator-like effector nucleases (TALENs), and the clustered regularly interspaced short palindromic repeats/associated protein (CRISPR/Cas) system. Because of its comparative advantages of higher editing efficiency, easier operation and lower cost [2], the CRISPR/Cas system has been shortlisted as one of the top ten technological breakthroughs by *Science* three times. As an adaptive immune system of bacteria and archaea, the CRISPR/Cas system is widely used in gene editing research due to the programmable characteristics of its guide RNA and the obvious activity of the Cas nuclease in a variety of cells and tissues [3]. French biochemist Charpentier, E. and American biologist Doudna, J.A. won the Nobel Prize in Chemistry in 2020 because of their outstanding contributions to the CRISPR/Cas9 gene editing technology, indicating the irreplaceable revolutionary significance of this system.

With the deepening of global biotechnology research and the vigorous development of the biological industry, the scientific achievements related to gene editing have continued to increase in recent years. Literature reviews on the technological advances and their applications in various fields have been conducted. In 2014, the working mechanism of the RNA-guided system in genome editing technology was first clearly summarized in eukaryotic cells [4]. Meanwhile, researchers summarized the genetic editing function of the CRISPR/Cas9 technology in mammalian cells and its application in cancer and other disease treatments [5]. When the engineered T cells have become mature, researchers introduced the CRISPR/Cas9 technology into the therapy of recurrent and refractory hematologic malignancies [6]. In addition, the CRISPR/Cas9 technology has also been widely used in crop breeding [7], including the development of fine-tuning gene regulation, traits improvement, virus resistance enhancement, etc. [8]. Recently, CRISPR has stepped forward rapidly fueled by the emergence of new tools such as base editors, transposases, and prime editors. Moreover, the box of Cas nucleases is expanding including Cpf1 and so on, and widely adopted in transcriptional regulation, epigenetic modifications, RNA editing, nucleic acid detection, etc. [9,10]. 

Most of the related reviews have summarized the gene editing characteristics, application scenarios, recent development trends of different tools, and optimization of gene editing tools. However, relatively few are about the objective, quantitative and comprehensive investigation of the literature related to gene editing [11,12]. Here, we use an integrated bibliometric approach [13,14] and visualization software such as Citespace and Scimago Graphica to better understand the research status, evolution disciplinarian, competition and cooperation situation, and development trend. The latest research frontiers and hotspots for the past two decades are also tracked on the basis of text clustering analysis. For this purpose, 13,980 articles and reviews published since 1999 are retrospectively collected from the Web of Science, and the trend of publications, authorship patterns, topics and biases in the field of gene editing are analyzed comprehensively. 

## 2. Materials and Method

The bibliometrics analysis is an important tool for comprehensive data mining and has been widely used due to its objectivity and quantitative characteristics. Peer-reviewed articles are obtained by text queries mining the Science Citation Index Expanded (SCIE) database in Web of Science, which is a natural scientific quotation index database of global authority, and currently includes more than 8800 international high-level journals in the field of natural sciences. Records were retrieved and cross-indexed by using entries that provide information with regard to manuscript authors, affiliated institutions, publication journals, years, titles and abstracts, spanning manuscripts published between 1999 and 2022 (up to 19 January 2022). The search strategy was based on the gene-editing-related queries (Appendix A). 

Data cleaning was conducted within the retrieved literature to limit the literature types to articles and reviews written in English, and finally narrowed down the results to 13,980 valid data (Appendix A). Literature was downloaded after the screening in the format of “summary, full record (including references),” saved as plain text files used for analyzing data samples, and finally imported to Citespace (Drexel University, Philadelphia, US), Scimago Graphica (SCImago Lab, Granada, Spain), and other visual software. By combining the documentary metrology analysis, visualization analysis, and social network analysis from three dimensions of the total amounts, citations and themes, we constructed the basic profiles and development dynamics of gene editing research.

## 3. Results

### 3.1. Origin and Eras of Gene Editing Technology

In 1953, the DNA double helix structure proposed by Watson and Crick realized the leap from general genetics to modern molecular biology and opened the prelude to gene editing technology [15]. The homologous recombination in yeast for the first time welcomed the era of transgene, which was promoted by the emergence of electroporation, PCR and other technologies. The first literature that explicitly proposed the concept of gene editing appeared in 1990, which is mainly about the RNA editing model of kinetoplastid. The partial hybrid formed between the gRNA and the pre-edited mRNA is a substrate that is relegated after multiple cycles of cleavage, addition, or deletion of uridylic acid, and finally produces a complete hybrid between the gRNA and the edited mRNA [16]. The successive discovery of meganuclease, ZFNs and TALENs in 1986 [17], 1996 [18], and 2010 [19], respectively, and their application in agriculture, medicine, and industry promoted the rapid transition from the era of transgene to the era of gene editing. These milestone events are listed in Figure 1. 

In particular, the CRISPR/Cas system has brought about a new round of revolution in gene editing technology [20] and led to a surge in gene editing-related research since 2014. The number of publications on CRISPR/Cas is several times more than that of the other three gene editing technologies with a relative wide application (Figure 2 and Table 1). The base editor and prime editor appeared in 2016 and 2019, respectively, and can be used to perform more precise gene editing with a single base, which is believed to open a new era of the CRISPR technology, along with the breakthrough and adoption of new CRISPR enzymes (Figure 1 and Figure 2). 

### 3.2. Development of Four Types of Gene Editing Technologies

Meganuclease is the earliest gene editing tool that recognizes DNA fragments in the range of 12–40 bp and causes double-strand break (DSB) [100]. Then, the genomic DNA is repaired mainly depending on non-homologous end joining (NHEJ) [101] (Figure 3). As the ORF is shown to be a site-specific DNA endonuclease that initiates homing events, Meganuclease is firstly used in genetic engineering [17]. Meganuclease can be used to treat human Xeroderma Pigmentosum by editing targeted gene *XPC* [21], to produce insect-resistant cotton in crop breeding [22], and to enhance *Phaeodactylum tricornutum*’s productivity of triglycerides by editing the UDP-glucose pyrophosphorylase gene [23] (Table 1). The low level of publication volume and limited application of the Meganuclease-mediated gene editing (Figure 2, Appendix A) are mainly due to two factors. First, hundreds of meganucleases have unique recognition sequences. It is thus necessary to find the very low likelihood of meganuclease targeting specific gene sequences [102]. Second, DSBs are mainly repaired by NHEJ, which cannot introduce foreign DNA templates and randomly delete or insert DNA fragments at the break sites [103].

ZFNs are the first sequence-specific nucleases that allow gene editing in living cells by inducing targeted DNA DSBs at specific genomic loci. They are composed of a zinc finger DNA binding domain responsible for specific recognition sequences (Cys2-His2) and a DNA cleavage domain for non-specific restriction endonuclease cleavage (*Fok*I endonuclease) [104]. Two independent zinc finger modules (composed of ZFPs) are designed to recognize the two strands of DNA, respectively, and then lead to DNA DSBs at the target site through *Fok*I-mediated homodimerization. After DNA damage, it is repaired by NHEJ and homology directed repair (HDR), thereby realizing the gene editing (Figure 3). With the improvement of the specificity of ZFNs as gene modification reagents [105], this technology has been gradually introduced and used in the agriculture and gene therapy areas. In 2005, Urnov, F. D. et al., performed targeted editing of the *IL2RG* gene in K562 and CD4+ T cells by ZFNs, making it possible to treat X-linked severe combined immune deficiency (SCID) by gene therapy [24]. In agriculture, ZFNs can be used to improve herbicide tolerance in maize [25]. Since many core patents of ZFNs technology are blocked by the Sangoma Company, ZFNs have not arrived at large-scale clinical application. Moreover, the limitations of ZFNs, such as being too dependent on the upstream and downstream sequences of the target gene, cytotoxicity, and high off-target rate [106,107,108], have greatly restricted the development and application of the technology (Figure 2, Appendix A). 

TALENs are transcription activator-like effector nucleases developed from effector TALEs in *Xanthomonas* [109]. TALENs are modular in form and function like ZFNs, consisting of a fusion of the *Fok*I cleavage domain and TALE protein DNA-binding domain [19]. When applying TALENs for gene editing, after a pair of TALENs are co-transfected into cells, two *Fok*Is form a dimer and then cleave DNA strands at the target site, and cause DSBs. Then, the cell repairs DNA through NHEJ or HDR, and indels are generated during the repair process, resulting in frame-shift mutations (Figure 3). Compared to ZFNs, the advantages of TALENs technology are the simpler design [110] and higher specificity [111], which lead to a wider application. Tesson, L. et al., used TALENs to disrupt the rat IgM locus, successfully inactivating the *IgM* gene and creating heritable mutations in the gene in a mammalian rat model [26]. Li, T. et al., developed dTALENs technology by targeting specific sites in the *URA3*, *LYS2*, and *ADE2* genes of *Saccharomyces cerevisiae* and found that *Saccharomyces cerevisiae* was undetectable for cytotoxicity and minimal levels of undesired mutations, which first extend the application of TALENs technology from mammalian cells to complete eukaryotes [28]. Targeted disruption of the rice bacterial blight susceptibility gene *Os11N3* (*OsSWEET14*) by TALENs enabled plants resistant to bacterial blight in rice, which has laid a preliminary research foundation for the application of TALENs in crop breeding [27]. However, this technology still has disadvantages such as cytotoxicity and the complex module assembly process [112]. The overall trend of publication volume of the TALENs technology is similar to that of ZFNs, and its publication volume also began to drastically decline after reaching its peak in 2016 (Figure 2, Appendix A).

The ZFNs and TALENs technologies have improved the efficiency of gene knockout, but new proteins need to be redesigned for different sites in the genome, requiring the cumbersome operation and high technical threshold. After a burst of efforts, in 2007, Barrangou, R. et al. demonstrated the type II CRISPR/Cas system derived from *Streptococcus thermophilus* had the function of the acquired immune system for the first time [113]. The CRISPR/Cas9 technology came out in 2012, using the DNA endonuclease Cas9 derived from bacteria instead of the *Fok*I restriction enzyme used by ZFNs and TALENs [114]. In 2012, Jinek, M. et al. first demonstrated that Cas9 could specifically cut the target DNA sequence under the guidance of synthetic sgRNA in vitro [20]. In 2013, Zhang, F.’s team published a related paper, demonstrating that CRISPR/Cas9 could be applied to the genome of human cells for the first time [29]. The simplified editing system consists of a single guide RNA and a Cas9 protein, and the Cas9-RNA complex cleave DNA through DSBs at the target site (Figure 3). Because the optimized system is simple, accurate, and fast, which greatly reduces the technical threshold and cost [115], CRISPR/Cas9 has become the most important tool for gene editing. Since 2014, the publication volume of CRISPR/Cas9 has rapidly increased and surpassed the other three gene editing technologies (Figure 2, Appendix A). The application of CRISPR/Cas9 is concentrated in the medical field, including animal models, gene therapy, and targeted drugs. Genome-wide CRISPR-based knockout screens can be used for functional genomics studies [116]. This technique can detect the genomic loci of cellular drug resistance [117], define how cells induce host immune responses, and how certain viruses induce cell death [118]. The non-functional elements discovered by gene editing technology provide a new method for studying the structure and evolution of the human genome and screening for drug targets [119]. Niu, Y. et al. successfully achieved a double knockout of *Ppar-**γ* and *Rag1* using the CRISPR/Cas9 system in 2014 and obtained the first Cas9-mediated knockout cynomolgus monkey [120]. Subsequently, by constructing a mouse model, Long, C. et al. targeted mutation of *DMD*, the main control gene of Duchenne muscular dystrophy [31] and obtained an ideal therapeutic effect [121,122]. In addition, CRISPR/Cas9 has made remarkable progress in the gene therapy of human hereditary tyrosinemia [123], hemophilia [124], intestinal neoplasia [36,37], lung adenocarcinoma [38], cataracts [39], obesity [30] and other diseases. 

CRISPR/Cas9 can also be used to improve animal and plant breeding in agricultural production. It functions well in improving plant breeding in terms of yield, resistance, and quality [32,33]. In 2013, the Gao, C.X. team successfully applied CRISPR/Cas9 to rice and wheat for the first time and achieved targeted mutations of *OsPDS*, *OsBADH2*, *OsMPK2* and *TaMLO* genes [34]. Wang, C. et al. edited the *OsLOGL5* gene through CRISPR/Cas9, which makes rice achieve a significant increase in the seed setting rate, the total number of grains, number of full grains per panicle, and 1000-grain weight [40]. For crop resistance, Shimatani, Z. et al. enhanced the herbicide tolerance of rice by editing rice *OsALS* and *OsFTIP1e* genes [41]. Jia, H. et al., used CRISPR/Cas9 technology to modify the citrus susceptibility gene *CsLOB1*, resulting in citrus resistance to citrus canker [70]. For crop nutrition and quality, taking tomato as an example, CRISPR/Cas9 has successfully been used to improve fruit size and nutritional quality [50,57], the regulation of maturity [61], the storage stability [51] and cultivation of high-quality seedless fruit [59]. However, the application of CRISPR/Cas9 in industry is as few as the other technologies mentioned above. Denby, C. M. et al. biosynthesized aromatic monoterpenes in yeast by targeting mutation of the *LIS* gene, making the Beer have a hoppy taste [79]. Tong, Y. et al. developed a CRISPR system for actinomycete genomes that can efficiently and reversibly control target genes *actIORF1* and *actVB* [80]. Cai, P. et al., edited *Pichia pastoris* genome and found that fatty alcohol production can be increased to 380 mg/L [81].

Then, a series of simpler, more accurate Cas enzymes have been found, and are used as alternatives of Cas 9, especially after the structural analysis has been completed [125]. CRISPR/Cas12a (Cpf1) [126], CRISPR/Cas13a (C2c2) [127], CRISPR/Cas13b (C2c6) [128], and CRISPR/Cas13d [129] were developed in 2015, 2016, 2017, and 2018, respectively, making a huge contribution to the development of GE technology. In 2018, Doudna, J.A.’s team successfully applied Cpf1 (Cas12a) to identify viral DNA [130]. Subsequently, it was used for SARS-CoV-2 virus detection of COVID-19 [83]. Cpf1 can improve soybean fatty acid content by targeting mutations in soybean fatty acid desaturases genes *FAD2-1A* and *FAD2-1B* [84]. *Corynebacterium glutamicum* is an important industrial bacteria, and Cpf1 successfully edited its genome for the first time [85]. CRISPR/Cas13a, CRISPR/Cas13b and CRISPR/Cas13d have been used for viral nucleic acid detection and genome editing of Dengue and Zika virus [94], SARS-CoV-1 [96] and SARS-CoV-2 [97], respectively. CRISPR/Cas13d has achieved gene editing of *Nicotiana benthamiana* and *Arabidopsis* thaliana RNA virus [98]. Another breakthrough in CRISPR is base editor (BE) [131]. Adenine base editors (ABEs) [35], prime editors (PEs) [132], and DddA-derived cytosine base editors (DdCBEs) [133] were invented by the Liu, D.R. team in 2017, 2019, and 2020, respectively, and have become major technology systems in the field of gene editing. In addition, Target-AID technology, dCas9-AIDx technology, and other single-base gene editing systems have been invented successively [93,134], with a relatively narrow application, especially in the fields of medicine and agriculture (Table 1).

Collectively, gene editing technology has a wide range of research and applications in agriculture, medicine, and industry (Figure 2, Appendix A). Among the three fields, it has the most research in health and medicine, where the publication number has been ranked first in the past two decades. The annual publication volume in the agriculture area is relatively lower, but the growth rate is higher. Studies of gene editing related to the industrial field are relatively few (Figure 2, Appendix A). In summary, the development and application trends of gene editing technology are simpler, safer, more efficient and more precise.

### 3.3. Competition and Cooperation of Key Actors in Gene Editing Technology

#### 3.3.1. National Performance

In general, the productivity and impact of the articles published by a certain country are two aspects to evaluate its competitiveness [1,2]. In the typical bibliometric analysis, the average citation is a robust index to quantify the influence of scientific research [3]. Accordingly, the United States is in the leading position, ranking first in publication volume and second in average citations. China ranks second in publication volume but 12^th^ in average citations. Comprehensively evaluating the total publication volume and the average citation frequency, we found that Germany, South Korea and France have great potential for development in gene editing. A special case is Sweden, which the total number of publications is only 208. However, the average citation frequency of Sweden has reached 84.28, which is 2.6 times higher than the average. Compared with Scotland and Russia, which have a similar number of publications, using the average citation frequency as an analysis indicator, it can be found that the influence of relevant research in Sweden is generally higher than that in Scotland and Russia, and Sweden is more authoritative in gene editing (Figure 4A, Appendix A).

The geographical distribution and cooperation of gene editing research are shown in Figure 4B. Not surprisingly, the United States is the largest contributor to scientific collaboration and has more frequent cooperative relationships with Japan, Germany, and the UK. China is next only to the United States, with USA and Germany as main partners. Russia, India, and Iran are the three countries with the least international cooperation. Collectively, the scientific research output of gene editing is mainly concentrated in the USA and several developed countries in Europe. Although China ranks second in publication volume, due to the complex factors including the late start, policy restrictions, and focusing more on agriculture, the overall scientific research contribution is still at a moderate level. The UK, Japan, and Germany are relatively important nodes in the innovation network. 

#### 3.3.2. Key Institutions and Scientists Engaged in Gene Editing Technology

The academic performance of the most productive research institutions and scientists is shown in Table 2 and Table 3, and Figure 5. The top five institutions with the largest publication volume are Chinese Academy of Sciences, University of Chinese Academy of Sciences, Harvard Medical School, Stanford University, and Harvard University, all of which are the top universities or institutes in their respective countries. In addition, Massachusetts Institute of Technology, along with Harvard University, Massachusetts Gen Hospital, and University of California Berkeley have relatively higher average citations, demonstrating their high-level academic output capability in gene editing. Interestingly, the four institutes constitute a “golden quadrangle” in the cooperative network (Figure 5). Another group of partners are Stanford University, University of Washington, and University of California, San Francisco. Chinese Academy of Sciences stands at the core of the cooperative relationship in China and has a strong correlation with University of California, Davis, and other institutions in the United States as well. 

The top five productive authors are Yamamoto, T.; Sakuma, T.; Zhang, F.; Kim, J.; and Doudna, J.A. Authors with the highest average citations are Zhang, F.; Joung, J. K.; Gregory, P. D.; Doudna, J.A.; and Michael C.H. According to the Recent 3-year rate, Qi, Y.P.; Zhang, Y.; and Gao, C.X. are quite active recently (Table 3). Specifically, Yamamoto, T. and Sakuma, T. have published the most papers in a relative early period and are closely collaborative scientists from Hiroshima University. Their main research direction is focused on the development and applications in organisms and cells of TALEN and CRISPR technologies [135,136]. Doudna, J.A. and Charpentier, E. shared the 2020 Nobel Prize in Chemistry for their discovery of the CRISPR/Cas9 genetic scissors, but the latter does not enter the top 20 researchers in terms of publication volume. Therefore, this shows that the number of publications of researchers cannot be used as the only reference factor to measure their contribution to a field. Breakthroughs in science often come from the integration of disciplines. Therefore, when cooperating with scholars in a certain field, their academic capabilities should be comprehensively considered, rather than only the number of publications as unique credentials. As one of the pioneer researchers to apply CRISPR/Cas to eukaryotic and human cells, Zhang, F. first reported and revealed the functional mechanism of the novel CRISPR system Cpf1 in 2015 [126] and has conducted a lot of research work to improve the efficiency of CRISPR/Cas system in recent years [137]. He is also the key researcher in the cooperative network. Kim, J. is an expert in the development of off-target effect evaluation tools for CRISPR/Cas. Gao, C.X. is the sixth most productive scientist, with an activity rate of 58% in the past three years, has paid more attention to the research and application of CRISPR/Cas in agriculture, and has a high academic output and active cooperation with other scientists, e.g., with Liu, D.R. to adapt prime editors in plants [90]. Barrangou, R. started researching the role of CRISPR/Cas in prokaryotic immunity in 2005. Liu, D.R. developed cytosine and adenine base editors for editing single nucleotides and developed prime editing, which can replace target DNA fragments with specific base sequences, thus keeping him at an emerging location of the global cooperative relationship. In general, China still lags behind the USA and some European countries in the number and academic influence of high-level scholars in the field of gene editing (Figure 5B). 

### 3.4. Research Hotspots and Evolution of Gene Editing Technology

The co-occurrence analysis of the keywords can reflect the key knowledge nodes, structure, and hot topics in the field of gene editing research. The six keywords with an occurrence frequency greater than 1000, are CRISPR/Cas9, expression, gene, DNA, genome, and mutation (Figure 6A, Appendix A), indicating that the CRISPR/ Cas9 technology holds an important position in the field of gene editing research and has been widely used for gene mutation and gene expression.

Figure 6B shows the top 20 keywords with the strongest citation bursts, illustrating the emergence and replacement of research hotspots from 1999 to 2022. According to the emergent words, gene editing has been mainly applied to the research of mammalian embryonic stem cells since 2005. Then, the focus shifted to molecular biology at the gene level due to the emergence of homologous recombination, double-strand break, DNA binding specificity, and gene targeting from 2006 to 2009. The emergence of ZFNs technology in 1996 triggered a new round of research upsurge. With further improvement and maturity, gene editing technology began to be applied to human cell research in 2014. In the past two years, the research hotspots of gene editing have tended to be genomic DNA and base editing. The realization of precise base editing technology benefits from the cytosine base editor, adenine base editor, and prime editing developed by Liu, D.R.’s laboratory [132,138].

Based on the text clustering analysis, a total of eight clusters are obtained, namely, #0, CRISPR; #1, homologous recombination; #2, gene expression; #3, identification; #4, mouse model; #6, sequence correction; #7, somatic cell; #8, noncoding RNA (Figure 7). It indicates that the application of gene editing in genetics, medicine, and gene therapy has become a focus of research in recent years. In addition, many research topics have shown a spurt in growth since 2012. The reason behind this may be the breakthrough of CRISPR/Cas9 which has made gene editing free from technical and cost limitations and greatly promoted the breadth and depth of related research. Correspondingly, the CRISPR is a clustering label word with the longest period from 2005 to 2022.

Regulation is an important but controversial topic for GE technology commercialization and has intrigued researchers and governments for decades (Figure 8, Appendix A). Since CRISPR/Cas9 launched in 2012, the boom of GE research and rapidly expanding application have raised a series of social ethics and food safety issues as well as economic and environmental benefits. Therefore, research publications related to the regulatory framework soar up with an exponential growth afterward (Figure 8). However, the growth rate of publications volume related to GE regulation has decreased significantly since 2019, reflecting the international trend of relaxation in GE regulation. In March 2018, the USDA clarified that GE crops were not being used as plant pests would not be regulated. As a result, the commercialization of GE crops has been accelerated, and the number of publications on regulatory research has thus declined.

According to Figure 9 (Appendix A), the keywords in regulatory research of GE are targeted mutagenesis, DNA, homologous recombination, protein, RNA, human cell, and so on, indicating that the focus of gene editing supervision is not only the technical details at the molecular level but also ethical issues of GE technology applied to human embryos and germ cells. However, keywords of regulatory such as off-target effects, efficiency or safety are missing or weak (Figure 9, Appendix A). We believe that due to the rapid development of CRISPR technology and its high technological maturity, compared with transgenic technology and other gene editing technologies, it has the advantages of accurate targeting, low off-target rate, low cytotoxicity, etc. Therefore, in the current regulatory process of CRISPR-based gene editing technology, compared with related research on transgenic regulation, it will weaken the aspects of off-target effects, efficiency and safety, and pay more attention to technical and ethical regulatory issues. For instance, gene-edited crops are theoretically safer and have better economic prospects. Some countries, represented by the United States, regard gene-edited crops as equivalent to naturally mutated plants, so they are not included in the strict supervision of genetically modified crops. At present, many countries have adopted laws or industry norms to prohibit genetic modification of germ cells, explicitly the gene editing of human embryos, while other countries still hold unclear attitudes. The Chinese academic community have studied and discussed the ethical issues involved in the application of GE technology, but there are still a lot of gaps in regulatory practice. Considering that the commercialization of GE crops is advancing irresistibly, the regulation of GE crops draws more attention, as well as whether regulation of GE crops in accordance with the standards of GMO has become the focus of controversy. The insertion of foreign genes and the existence of off-target effects are the main risks followed by the regulatory strategies in various countries.

As from Figure 10, the USA has the most publications and the strongest intermediary centrality in the field of GE regulation, indicating its leading position and bridging role. China ranks second in terms of publications and centrality, while the regulatory framework is still not certain, and thus a hot point of contention in China. The following countries are Germany, Japan, and the UK, with great development research potential and influence. Although Sweden, South Korea, and Italy have published fewer papers on gene editing regulation, their international cooperation is very active (Figure 10, Appendix A).

In the process of formulating and implementing GE regulations, the management ideas and practices in various countries are quite different. The USA, Canada, and other countries focus on product supervision, the EU focuses on process supervision, and China’s supervision includes both process and product supervision. Internationally, there is no clear consensus on whether gene-edited organisms are subject to regulations as GMOs. Taking GE crops as examples, the EU considers crops that have undergone the transfer of exogenous DNA to be GMOs even if the final product does not contain exogenous DNA. The USA, Canada, Australia, Japan, Argentina, and some other countries treat GE crops without inserted nucleic acids as non-GMO and allow them to enter the market [139,140,141]. In contrast, China’s GE regulatory policies are more conservative than those of the USA, but looser than the EU’s tough stance on GE crops. The first time that the GE concept been separated from GMO in the regulative level by the Chinese Ministry of Agriculture and Rural Affairs on 24 January 2022, will greatly facilitate research on GE crops in China to cultivate agricultural plant species with higher yields, greater disease resistance, and resilience to climate change [142].

## 4. Discussion and Perspectives

Based on the bibliometric evaluation of gene editing technology and its application in subdivisions over the last two decades, we identified the competitiveness and cooperation among main countries, institutions, and authors. Our data demonstrate the usefulness of Web of Science, in this sense, which can be applied in any area. The methodology can be further used to explore the popular topics and trends associated with GE research, combined with social network analysis. Will the authors with high “recent 3-year rate” be the intermediate forces in gene editing? How does the emerging keywords such as” lipid nanoparticle” function in the development of gene editing? Can base editor, prime editor or undiscovered Cas proteins be the future of CRISPR? Will a burst of gene editing innovation come from within the field or from collaboration between various disciplines? It is believed that bibliometrics never claims to offer insights into scientific knowledge, but it can put at the service of advancing scientific research in actual practice as a useful tool [143].

With this in mind, we selected and analyzed the high-quality and typical papers according to the quantitative result bibliometric analysis. By observing the academic pathways of the most active authors, and the scientific reasons for the changes in research trends, the main bottlenecks in the development and application of GE technology are inferred. The first issue involves the PAM sequences in the CRISPR/Cas technology. In practical applications, the PAM sequence of the traditional CRISPR/Cas9 system only recognizes NGG, restricting the selection of specific target sites. The second issue involves the low genetic transformation efficiency in some plants. For example, two transformation systems in plants: *Agrobacterium*-mediated and particle bombardment, have disadvantages such as multiple copies of DNA insertion, difficulties in enhancing genetic transformation efficiency in certain genetic backgrounds, etc. Recent research has shown that nanomaterials can transport Cas9 protein into cells to edit target genes, which may broaden the original transformation methods and accelerate the genetic improvement of crops [144]. However, there is still substantial work that needs to be done to improve operational efficiency. The third issue involves the off-target phenomenon. The CRISPR/Cas9 technology relies on the Cas9 nuclease to cleave the target DNA at a site 3-bp upstream of the PAM site under the guidance of gRNA (on-target), but also cleave non-target sites (similar to the sgRNA target site sequences and possessing the PAM sites), which results in so-called off-targets. In addition, the Cas9 recognizes not only standard PAMs but also non-standard PAMs, which may also cause a certain degree risk of off-targets, ultimately resulting in a series of uncontrollable mutations. Some studies have suggested that off-target CRISPR editing may increase the risk of cancer [145]. In a word, these limitations and problems of the gene editing technology must be overcome so that it can be used to better serve human health and better life, especially in the global spread of COVID-19 and similar living environments that humans need to face in the long future.

The latest keywords in the timeline and achievements of the leading authors may chart a course for the future of a certain area. In this sense, future strategies to promote the development of gene editing technology may include three aspects. The first aspect involves engineering Cas proteins and optimizing sgRNAs to reduce off-target effects as well as developing more accurate detection methods of off-targets. To weaken the restriction of Cas9 protein by PAM, the Cas9 protein can be modified so that more orthologous enzymes of Cas9 can be used in gene editing systems. For example, Akcakaya, P. et al. [146] reported that gRNA plays a key role in circumventing the off-target effects of CRISPR/Cas. They developed a method for verification of in vivo off-targets (VIVO), which can stably identify the off-target effects of CRISPR/Cas in the whole genome in vivo. The base editor can complete single-base substitution without DBS. The CRISPR variant (prime editor) formed by the fusion of catalyzed damaged Cas9 nuclease and reverse transcriptase does not require a donor DNA template. In this way, precisely targeted insertion, deletion, and point mutation can be achieved, which will also be an important development direction of gene editing and its application in the future. In addition, trying new transfection methods such as nuclear injection of sgRNA and Cas9 complexes and establishing stable cell lines may alleviate the problem of low transformation efficiency. Secondly, effectively expanding the range of CRISPR/Cas9 nuclease PAM sites is the key to the widespread application of the CRISPR/Cas system in gene editing. In 2018, Nishimasu, H. et al. obtained a spCas9 variant (spCas9-NG) with a PAM sequence of NG by modifying the sequence of spCas9 [147], breaking through the limitation of the PAM sequence and significantly expanding the candidate range of target sites. The third aspect pertains to developing new types of nucleases and new mRNA delivery technologies. Zhang, F.’s team published consecutive papers in *Science* in 2021, discovering a class of proteins called IscB in bacteria. The protein IscB retains programmable features and no other redundant structures, making it easier to deliver into the body, and allows scientists to add more new functions to this class of proteins through protein engineering, including PEG10-based human proteins [148,149,150]. These relevant results indicate that there may be many undiscovered proteins such as Cas12b/C2c1 [124,151] with similar functions to Cas9 nuclease in nature, which means that more research tools, drug delivery methods, and treatment modes are expected to be explored and developed in the future. 

To date, the USA has granted GMO regulatory exemptions for GE canola, high-oleic soybeans, antioxidant mushrooms, waxy corn, and some other crops, and regulates most GE crops as conventional plants. Japan, Finland, Sweden, Russia, Brazil, Argentina, and many other countries have also listed some GE plant products as non-GMO products. China has issued the *Guidelines for Safety Evaluation of Agricultural Gene Editing Plants (Trial Edition)*, in which it is shown that GE products without externally processed nucleic acids will not be treated as GMOs. Nowadays, it is widely accepted that the GE products should be handled differently according to whether or not they use the repair template and the types of targeted sequence changes. Collectively, there is a trend of policy consistency all over the world, and many countries tend to carry out the de-regulation of GE products. In the future, as more countries update their regulatory frameworks on gene editing, more GE products will flow into the market to improve the quality of life.

## Figures and Tables

**Figure 1 cells-11-02682-f001:**
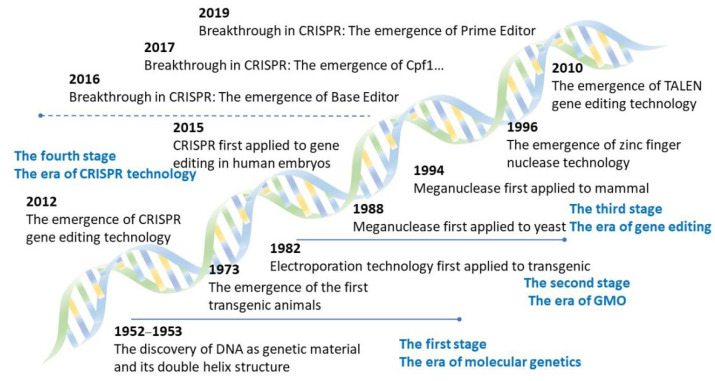
The development history of general genetics, modern molecular biology, GMO and GE.

**Figure 2 cells-11-02682-f002:**
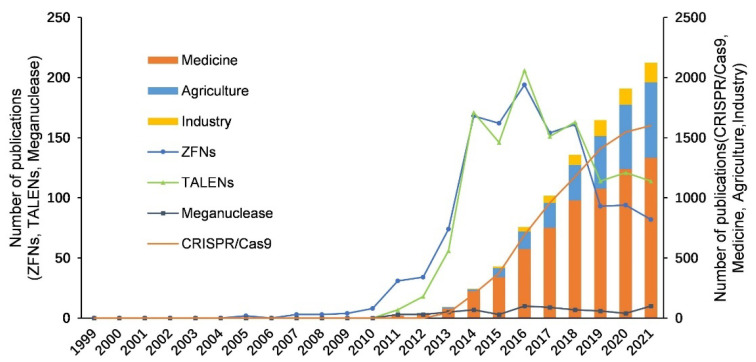
GE-related publications from 1999 to 2021. The number of publications in three sectors (medicine, agriculture and industry) and four different GE technologies.

**Figure 3 cells-11-02682-f003:**
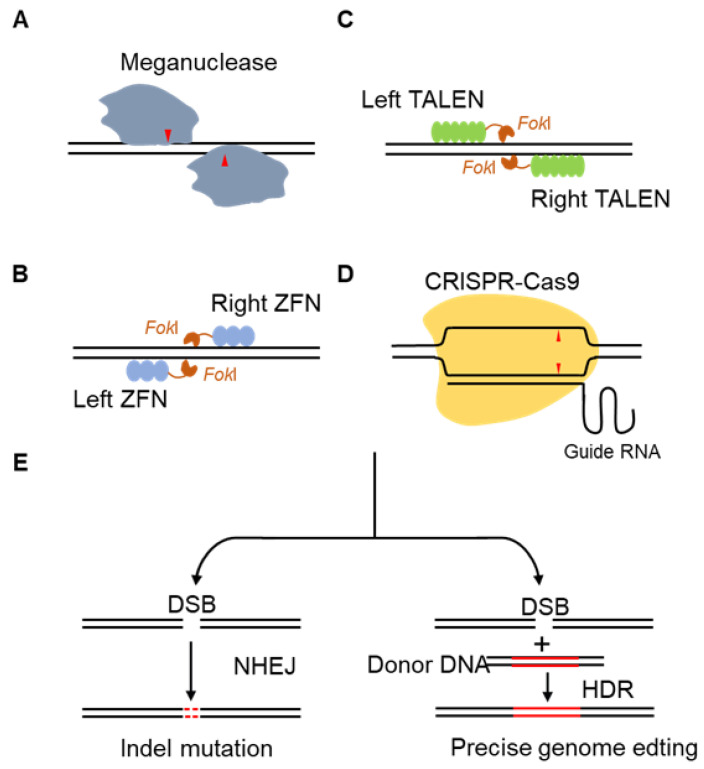
Schematic diagram of four types of gene editing technologies. (**A**) Schematic diagram of Meganuclease editing genomic DNA. (**B**) Schematic diagram of zinc-finger nuclease (ZFN) editing genomic DNA. (**C**) Schematic diagram of transcription activator-like effector nuclease (TALEN) editing genomic DNA. (**D**) Schematic diagram of clustered regularly interspaced short palindromic repeats (CRISPR)/Cas9 editing genomic DNA. (**E**) Schematic diagram of two DNA repair pathways (NHEJ and HDR) when all four enzymes edit and produce DNA double strand breaks (DSBs).

**Figure 4 cells-11-02682-f004:**
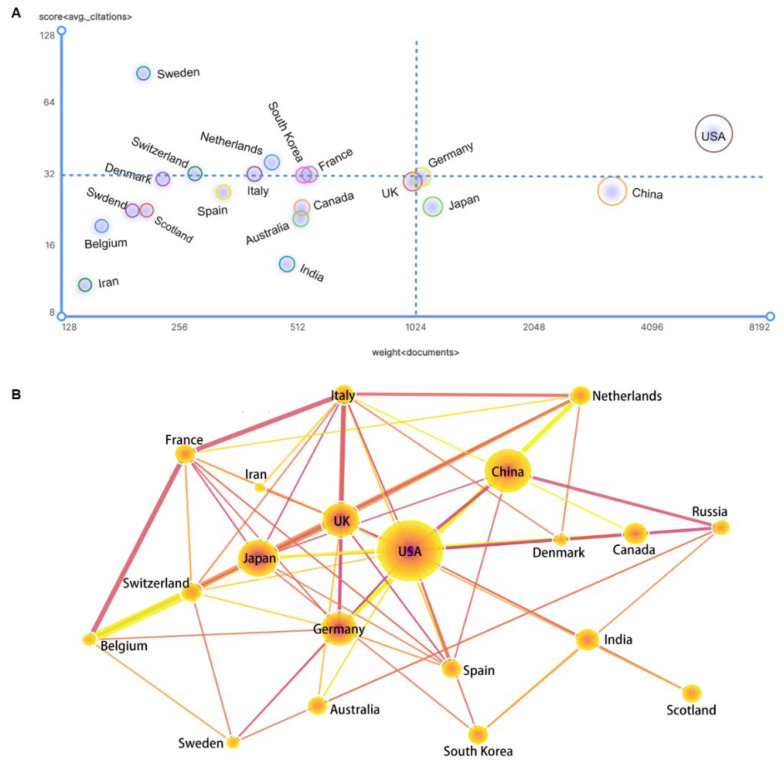
The academic performance of the top 20 countries. (**A**) Bubble chart of publications and average citations in the 20 countries. (**B**) National cooperation network among the 20 countries.

**Figure 5 cells-11-02682-f005:**
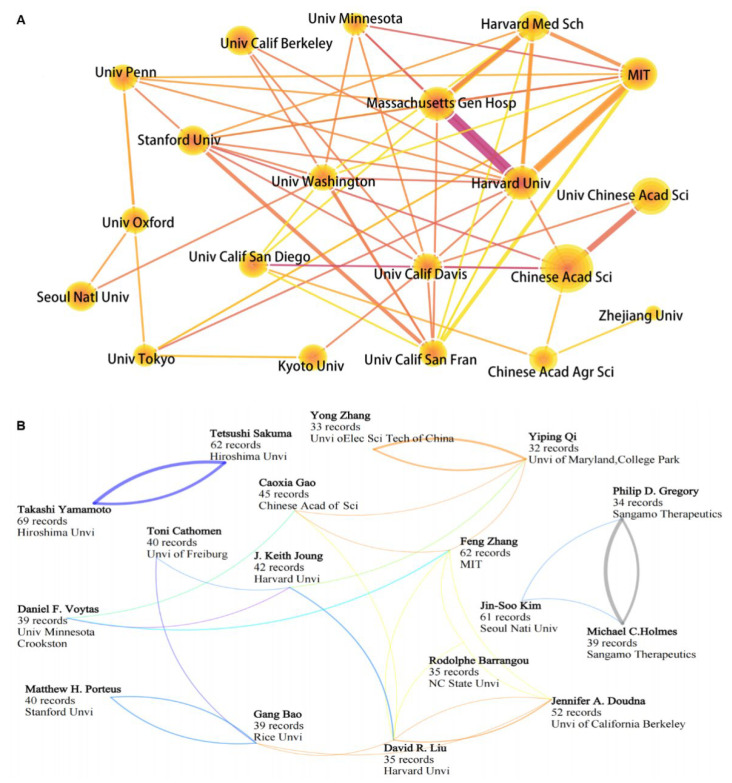
The academic performance of key institutes and researchers. (**A**) Cooperation network of the top 20 institutes. (**B**) Map of the collaborative patterns among the top 20 prolific scientists.

**Figure 6 cells-11-02682-f006:**
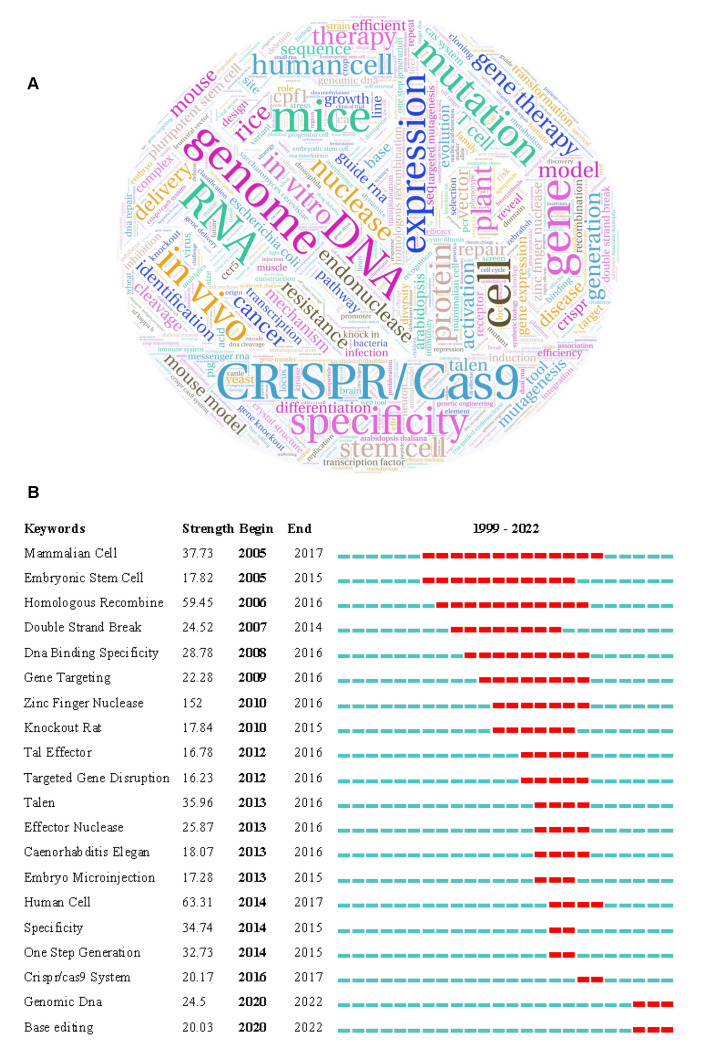
The hotspots of GE research. (**A**) The word cloud map of keywords according to the frequency of occurrence. (**B**) The emergence graph of the top 20 keywords.

**Figure 7 cells-11-02682-f007:**
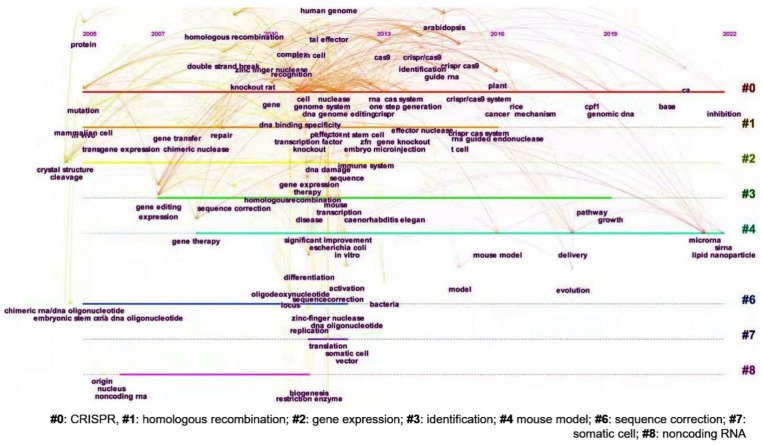
The keyword cluster timing diagram with the change of keywords from 2005 to 2022.3.5 The Worldwide Regulatory Framework.

**Figure 8 cells-11-02682-f008:**
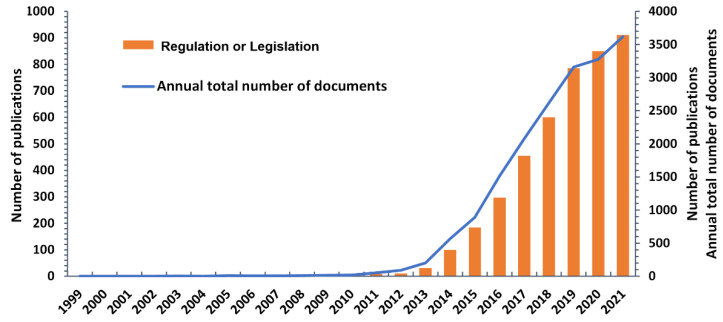
Regulation-related publications of GE from 1999 to 2021.

**Figure 9 cells-11-02682-f009:**
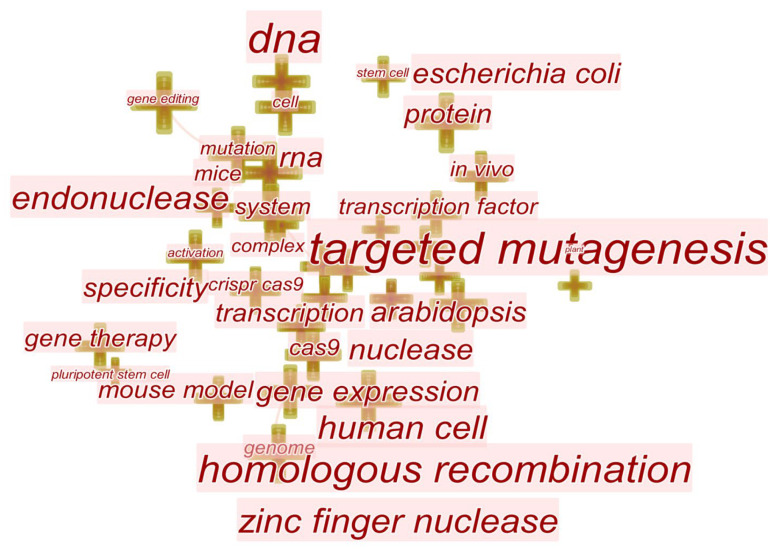
Keyword co-occurrence network diagram of GE regulation-related research.

**Figure 10 cells-11-02682-f010:**
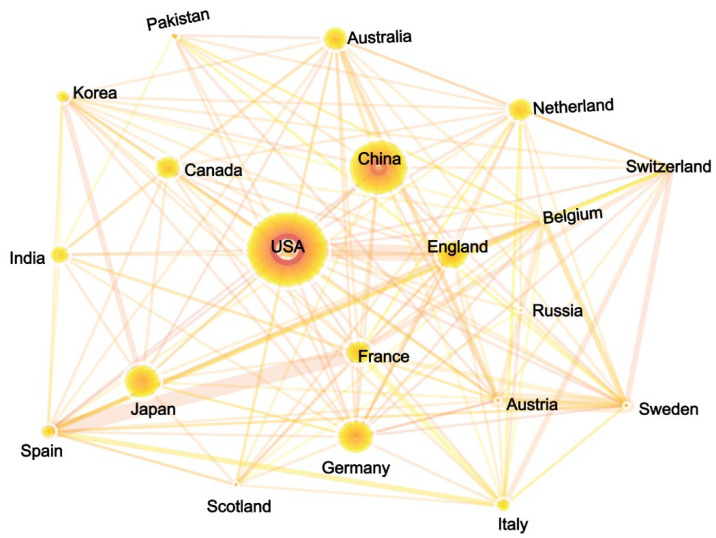
The co-occurrence network analysis of national cooperation in GE regulatory research.

**Table 1 cells-11-02682-t001:** The research status and application fields of GE technologies (mainly CRISPR/Cas9).

GE Technologies	Research Status				Typical Application Cases			
	Creators	Institutions and Countries	Years	Refs	Application Fields	Research Objects	Target Genes	Refs
Meganuclease	Colleaux. L	CNRS Laboratoire and Université Pierre et Marie Curie (France)	1986	[17]	Medicine and Health	Human xeroderma pigmentosum	*XPC*	[21]
					Agriculture	Cotton insect resistant	*hppd* and *epsps*	[22]
					Industry	*Phaeodactylum tricornutum*’s productivity of triglycerides	UDP-glucose pyrophosphorylase gene	[23]
ZFNs	Srinivasan Chandrasegaran	Johns Hopkins University (JHU, USA)	1996	[18]	Medicine and Health	K562, CD4^+^ T cells (X-linked severe combined immune deficiency (SCID))	*IL2RG*	[24]
					Agriculture	Maize herbicide tolerance	*Ipk1* and *Zp15*	[25]
TALENs	Daniel F. Voytas	University of Minnesota (USA)	2010	[19]	Medicine and Health	Rat immunoglobulinM (rat model)	*IgM*	[26]
					Agriculture	Rice disease-resistant (*Xanthomonas oryzae*)	*Os11N3*	[27]
					Industry	*Saccharomyces cerevisiae*	*URA3, LYS2* and *ADE2*	[28]
CRISPR/Cas9	Jennifer A. Doudna and Emmanuelle Charpentier	Howard Hughes Medical Institute (HHMI, USA) and The Laboratory for Molecular Infection Medicine Sweden (MIMS, Sweden)	2012	[20]	Medicine and Health	Human 293FT and mouse cells (first mammalian model)	*SpCas9*, *SpRNase III*, *EMX1*, *PVALB* and *Th*	[29]
						Cynomolgus monkey (mammalian model)	*Nr0b1, Ppar-γ and Rag1*	[30]
						Duchenne muscular dystrophy(mouse model)	*DMD*	[31,32,33]
						Human hereditary tyrosinemia (mouse model)	*Fah*	[34]
						Human hemophilia (mouse model)	*Serpinc1*	[35]
						Human intestinal neoplasia (mouse model)	*APC, P53* (*TP53*)*, KRAS* and *SMAD4*	[36,37]
						Human lung adenocarcinoma (mouse model)	*KRAS, p53 and LKB1*	[38]
						Cataracts (mouse model)	*Crygc*	[39]
						Human obesity (mouse model)	*FTO*	[30]
						Resistance to *Clostridium septicum* alpha-toxin or 6-thioguanine (mouse ES cells)	Whole genome	[40]
						Functional genomics in human cells	Anthrax and diphtheria toxin host genes	[41]
						Human autologous CD34+ cells	*BCL11A*	[42]
						Human hepatocytes	*TTR*	[43]
						Human fibroblast	*APP*	[44]
						Mouse acute myeloid leukemia cell	*RPA3,Brd4, Smarca4, Eed, Suz12* and *Rnf20*	[45]
						204 human cancer cell lines	*18,009 genes*	[46]
						SARS-CoV-2	*SARS-CoV-2 N1/N2/N3, pH1N1 H1* and *pH1N1/H275Y N1*	[47]
						Human leukemic	*BCR-ABL*	[48]
						Human HeLa cells	Telomerase gene	[49]
					Agriculture	Rice and Wheat (first applied to plants)	*OsPDS*, *OsBADH2*, *OsMPK2* and *TaMLO*	[34]
						Rice herbicide tolerance	*OsALS* and Os*FTIP1e*	[41]
						Tomato fruit size, quantity and nutritional value	*SELF-PRUNING, OVATE, FASCIATED, FRUIT WEIGHT 2.2, MULTIFLORA and LYCOPENE BETA CYCLASE*	[50]
						Tomato storability	*SP5G*	[51]
						Wheat grain weight enhancement	*TaGASR7*	[52]
						Sativa oleic acid content	*FAD2*	[53]
						*Brassica napus* L. (yellow seed traits, and increase protein content and fatty acid content)	*BnTT8*	[54]
						Oil content of rapeseed	*BnSFAR4 and BnSFAR5*	[55]
						Rice grain quality	*Waxy*	[56]
						Tomato fruit nutrition	*GABA-T/SSADH*	[57]
						Rice amylose	*Wx*	[58]
						Tomato high quality seedless fruit	*SlAGL6*	[59]
						Mitochondrial function and fruit ripening in tomato	*SlORRM4*	[60]
						Tomato ripening regulation	*lncRNA1459*	[61]
						Grape resistance to *Botrytis cinerea*	*VvWRKY52*	[62]
						Citrus (improvement of citrus canker resistance)	*CsLOB1*	[63]
						Wheat resistance to mildew	*TaEDR1*	[64]
						Rice resistance to salinity tolerance	*OsRR22*	[65]
						Rice herbicide-tolerant	*OsALS1*	[66]
						Potato resistance to the herbicide	*ALS1*	[67]
						Rice disease resistance	*OsSWEET13 and ten EBEPthXo2*	[68]
						Rice broad-spectrum disease resistance	*Bsr-k1*	[69]
						Duncan grapefruit (disease resistant citrus varieties)	*CsLOB1*	[70]
						Rice resistance to plant hoppers and stem borers	*OsCYP71A1*	[8]
						Rice tolerance to abiotic stresses	*OsSRFP1*	[71]
						Rice seed setting rate, the total number of grains, number of full grains per panicle and 1000-grain weight	*OsLOGL5*	[40]
						Rice grain size, width and weight	*OsGS3, OsGW2* and *OsGn1a*	[72]
						Rice ear length, grain size, cold tolerance	*OsPIN5b, GS3* and *OsMYB30*	[73]
						Rice grain number, ear structure, particle size	*Gn1a, DEPT, GS3* and *IPA1*	[74]
						Soybean NSPP and leaflet shape	*GmJAG1*	[75]
						Maize grain yield	*ARGOS8-v1, ARGOS8-v2* and *Zmcle7*	[76,77]
						Wheat grain shape and weight	*TaGW7*	[78]
					Industry	Yeast biosynthesizing monoterpenes	*LIS*	[79]
						*Actinomycetes*	*actIORF1* and *actVB*	[80]
						*Pichia pastoris*	*RAD52*	[81]
CRISPR /Cas12a(Cpf1)	Feng Zhang	Broad Institute of MIT and Harvard (USA)	2015	[76]	Medicine and Health	First mammalian model (HEK 293T cell)	*DNMT1*, *EMX1*, *VEGFA* and *GRIN2*	[82]
						SARS-CoV-2	SARS-CoV-2 N gene, E gene	[83]
					Agriculture	Soybean fatty acid desaturases	*FAD2-1A and FAD2-1B*	[84]
					Industry	*Corynebacterium glutamicum* genome	*crtYf* and *recT*	[85]
Base Editor (BE)	David R. Liu	Harvard University (USA)	2016	[86]	Medicine and Health	Albinism (mouse model)	*Dmd* and *Tyr*	[87]
					Agriculture	Rice, Wheat and Maize genome	*OsCDC48, OsNRT1.1B*, *OsSPL14*, *TaLOX2* and *ZmCENH3*	[88]
					Industry	*Escherichia coli* and *Brucella melitensis* genome	*rppH and lacZ*	[86]
Adenine Base Editors (ABEs)	David R. Liu	Harvard University (USA)	2017	[89]	Medicine and Health	Duchenne muscular dystrophy (mouse model)	*Tyr*	[89]
Prime Editors (PEs)	David R. Liu	Harvard University (USA)	2019	[90]	Agriculture	Rice and Wheat genome	*OsALS*, *OsCDC48*, *OsDEP1*, *OsEPSPS*, *OsGAPDH*, *OsLDMAR*, *TaGW2, TaUbi10, TaLOX2, TaMLO, TaDME* and *TaGASR7*	[90]
DddA-derived cytosine Base Editors (DdCBEs)	David R. Liu	Harvard University (USA)	2020	[91]	Medicine and Health	Human mitochondrial (HEK 293T cell)	*ND1, ND4, ND5.2, COX3.1* and *RNR2*	[91]
Target-AID	Akihiko Kondo	Kobe University (Japan)	2016	[92]	Agriculture	Tomato hormone	*DELLA and ETR1*	[41]
					Industry	*Escherichia coli* genome	*galK* and *rpoB*	[92]
dCas9-AIDx	Yan Song and Xing Chang	Chinese Academy of Sciences and Shanghai Jiao Tong University School of Medicine (China)	2016	[93]	Medicine and Health	Chronic myeloid leukemia cells	*GFP*	[93]
CRISPR /Cas13a (C2c2)	Feng Zhang	Broad Institute of MIT and Harvard (USA)	2016	[77]	Medicine and Health	Dengue and Zika virus	the *P. aeruginosa* acyltransferase gene and the *S. aureus* thermonuclease gene	[94]
						Human monocytic cell	*GBA1*	[95]
CRISPR /Cas13b (C2c6)	Feng Zhang	Broad Institute of MIT and Harvard (USA)	2017	[78]	Medicine and Health	SARS-CoV-1	RNA genome	[96]
CRISPR /Cas13d	Feng Zhang	Broad Institute of MIT and Harvard (USA)	2018	[79]	Medicine and Health	SARS-CoV-2	RNA genome	[97]
					Agriculture	*Nicotiana benthamiana* and *Arabidopsis thaliana* RNA virus	*GFP*	[98]
					Industry	*Ruminococcus flavefaciens* (frontotemporal dementia)	*ANXA4*	[99]

**Table 2 cells-11-02682-t002:** The top 20 institutions in gene editing with the publication volume as the first rank.

Rank	Organizations	Number of Records	Average Citations
1	Chinese Academy of Sciences	599	37.53
2	University of Chinese Academy of Sciences	311	30.90
3	Harvard Medical School	291	40.30
4	Stanford University	261	46.75
5	Harvard University	251	280.71
6	University of California, Berkeley	220	118.13
7	Chinese Academy of Agricultural Sciences	192	22.41
8	Massachusetts Institute of Technology	173	298.36
9	University of California, San Diego	160	39.74
10	University of California, San Francisco	155	54.07
11	Seoul National University	153	70.65
12	The University of Tokyo	152	36.21
13	University of Oxford	151	22.07
14	University of Minnesota	148	46.14
15	University of Pennsylvania	147	56.56
16	Zhejiang University	146	20.82
17	University of Washington	144	40.56
18	Kyoto University	138	24.96
19	Massachusetts Gen Hospital	137	173.14
20	University of California, Davis	136	27.55

**Table 3 cells-11-02682-t003:** The top 20 prolific authors in gene editing with the publication volume as the first rank. The recent 3-years rate is the percentage of publications published in the recent three years as a proportion of the author’s total publications.

Author	Number of Records	Total Citations	Average Citations	Year Range	Recent 3-Year Rate	Organizations	Top Research Fields (Number of Records)
Takashi Yamamoto	69	1866	27.04	2012–2021	20%	Hiroshima University	Cell Biology (24); Multidisciplinary Sciences (21); Developmental Biology (13)
Tetsushi Sakuma	62	1763	28.44	2012–2021	19%	Hiroshima University	Cell Biology (22); Multidisciplinary Sciences (19); Developmental Biology (11)
Feng Zhang	62	36,524	589.1	2012–2021	25%	Massachusetts Institute of Technology	Biochemistry Molecular Biology (22); Cell Biology (21); Multidisciplinary Sciences (15)
Jin-Soo Kim	61	9354	153.34	2009–2021	24%	Seoul National University	Biotechnology Applied Microbiology (24); Biochemistry Molecular Biology (15); Multidisciplinary Sciences (14)
Jennifer A. Doudna	52	14,594	280.65	2012–2021	44%	University of California, Berkeley	Multidisciplinary Sciences (23); Biochemistry Molecular Biology (14); Cell Biology (7)
Caixia Gao	45	3965	88.11	2014–2021	58%	Chinese Academy of Sciences	Biotechnology Applied Microbiology (15); Plant Sciences (11); Biochemistry Molecular Biology (9)
J. Keith Joung	42	16,411	390.74	2010–2021	20%	Harvard University	Biotechnology Applied Microbiology (15); Biochemistry Molecular Biology (9); Multidisciplinary Sciences (8)
Matthew H. Porteus	40	2762	69.05	2013–2021	44%	Stanford University	Medicine Research Experimental (11); Multidisciplinary Sciences (9); Biotechnology Applied Microbiology (8)
Toni Cathomen	40	2033	50.83	2009–2021	28%	University of Freiburg	Medicine Research Experimental (15); Genetics Heredity (12); Biotechnology Applied Microbiology (11)
Gang Bao	39	5242	134.41	2013–2021	39%	Rice University	Medicine Research Experimental (12); Biotechnology Applied Microbiology (8); Genetics Heredity (7)
Daniel F. Voytas	39	4877	125.05	2011–2021	24%	University Minnesota Crookston	Plant Sciences (18); Biotechnology Applied Microbiology (10); Multidisciplinary Sciences (7)
Michael C. Holmes	39	9602	246.21	2007–2021	18%	Sangamo Therapeutics	Biotechnology Applied Microbiology (14); Medicine Research Experimental (13); Genetics Heredity (9)
David R. Liu	35	8268	236.23	2013–2021	38%	Harvard University	Multidisciplinary Sciences (17); Biochemistry Molecular Biology (8); Biotechnology Applied Microbiology (6);
Rodolphe Barrangou	35	1996	57.03	2013–2021	49%	North Carolina State University	Microbiology (10); Biochemistry Molecular Biology (8); Biotechnology Applied Microbiology (7)
Philip D. Gregory	34	10,762	316.53	2007–2018	31%	Sangamo Therapeutics	Biotechnology Applied Microbiology (9); Multidisciplinary Sciences (7); Biochemical Research Methods (5)
Yong Zhang	33	1654	50.12	2015–2021	60%	University of Electronic Science Technology of China	Plant Sciences (14); Biochemistry Molecular Biology (11); Biotechnology Applied Microbiology (8)
Bing Yang	33	2103	63.73	2011–2021	44%	Iowa State University	Plant Sciences (19); Biotechnology Applied Microbiology (12); Biochemistry Molecular Biology (10)
YiPing Qi	32	1682	52.56	2015–2021	69%	University of Maryland, College Park	Plant Sciences (25); Biotechnology Applied Microbiology (6); Biochemistry Molecular Biology (5)
Xingxu Huang	28	1447	51.68	2014–2021	32%	Shanghai Tech University	Biochemistry Molecular Biology (9); Cell Biology (9); Multidisciplinary Sciences (7)
Huimin Zhao	28	1618	57.79	2012–2021	31%	University of Illinois, Urbana-Champaign	Biotechnology Applied Microbiology (14); Biochemical Research Methods (6); Biochemistry Molecular Biology (3)

## Data Availability

All data are shown in the main manuscript and in the Appendix A.

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
