# Peer review of "The Bibliometric Landscape of Gene Editing Innovation and Regulation in the Worldwide"

_cells, 2022, doi:10.3390/cells11172682_

Round 1

Reviewer 1 Report (Previous Reviewer 1)

In my previous review of the present paper I asked for a minor modification of the Discussion section, which seems to have been correctly performed. Therefore, from my point of view the paper can be accepted.

I also noticed that the paper has been expanded, probably according to the suggestions of other referee(s). Additions included a new section on regulatory issues, which seems to me well done.

Author Response

Thanks for the kind suggestions and encouragement. All the revised and polished language expression are marked in revision mode in the revised version.

Reviewer 2 Report (New Reviewer)

The manuscript

The Bibliometric Landscape of Gene Editing Innovation and Regulation in the Worldwide

by Wei et al, submitted to Cells for publication focuses on the history, present and future of genome editing technologies, based on the analysis of bibliographic data including 13980 publications between 1999 and 2022.

I have read the manuscript with great expectation, and I think it will be of great interest for the readers of the Journal.

I have few suggestions on the way to improving this manuscript:

1. The authors need to ensure that all the abbreviations are explained at their first appearance.

2. Figure 1. It is misleading to date the emergence of ZFN technology at 2002, as the first ZFN was described in 1996. Later on P17 this date is already 2010 for some reason: "The emergence of ZFNs technology in 2010 triggered a new round of research upsurge."

3. P5. The authors state that "ZFNs are the first sequence-specific nucleases that allow gene editing in living cells by inducing targeted DNA DSBs at specific genomic loci, and are composed of zinc finger proteins (ZFPs) Cys2-His2 and a non-specific DNA restriction enzyme (FokI endonuclease) [28]."

However, this is not correct. FokI is a specific restriction endonuclease. Only the nuclease domain of this enzyme is used to construct the ZFNs (and TALENs), which is indeed, non-specific without the ZF or DNA TALE recognition domains.

4. Figure 4. The authors state: "Sweden, Netherlands, and France are potential 276 competitors in gene editing research with low publication volume but high quality”

I wonder why these countries have been selected to mention. I would say, that based on the data in the table e.g. Germany, South Corea, and Italy seem to have larger contribution than Switzerland.

It is not only the citation number that characterizes the quality of a publication. However, it can be one of the measures. Nevertheless, the comparison of the average number of citations can be misleading when the number of publications taken into account is very different. The average number of citations is very sensitive to the deviations when low number of publications is included, but almost completely insensitive when large number of publications are included in the evaluation. I would suggest to include the ranges of the citations as a kind of error bars in the figure. It could be also interesting to select e.g. the 100 most cited publications from each country for comparison in another figure panel.

5. When analysing the authors’ contributions the authors state: "Doudna and Charpentier 320 shared the 2020 Nobel Prize in Chemistry for their discovery of the CRISPR/Cas9 genetic 321 scissors, but the latter doesn’t enter the top 20 researchers in terms of publication volume."

This is an important finding and shows that the publication volume of an author does not necessarily reflect the real contribution to the field. The collaboration of researchers – sometimes even from very different research fields – may lead to new results. However, the collaboration partner shall not be selected based on the publication volume – which might partially be suggested by the manuscript.

6. P20. The authors state: "According to Figure 9 (Supplementary Table 7), the keywords in regulatory research of GE are targeted mutagenesis, DNA, homologous recombination, protein, RNA, human cell, and so on, indicating that the focus of gene editing supervision is not only the technical details at the molecular level, but also ethical issues of GE technology applied to human embryos and germ cells."

Unfortunately, the keywords like off-target effects or efficiency or safety are missing or weak in Figure 9 and Table S7. In my opinion it needs to be discussed whether sometimes the potential profit from using e.g. gene edited crops is considered to be more important than the safety, which can’t be assessed in a short term experiment.

7. The manuscript needs to be carefully checked for typos.

Author Response

The Bibliometric Landscape of Gene Editing Innovation and Regulation in the Worldwide by Wei et al, submitted to Cells for publication focuses on the history, present and future of genome editing technologies, based on the analysis of bibliographic data including 13980 publications between 1999 and 2022.

 I have read the manuscript with great expectation, and I think it will be of great interest for the readers of the Journal.

 I have few suggestions on the way to improving this manuscript:

1.The authors need to ensure that all the abbreviations are explained at their first appearance.

Response: Kind suggestions and accepted.

Revision explanations: As you suggested, we have checked all the abbreviations, and ensure they are explained at their first appearance.

  1. Figure 1. It is misleading to date the emergence of ZFN technology at 2002, as the first ZFN was described in 1996. Later on P17 this date is already 2010 for some reason: "The emergence of ZFNs technology in 2010 triggered a new round of research upsurge."

Response: Kind suggestions and accepted.

Revision explanations: As you suggested, we have revised the wrong year of the emergence of ZFN technology, which is not in 1996. Actually, the date is 2010 (Figure 1, Line 370).

  1. P5. The authors state that "ZFNs are the first sequence-specific nucleases that allow gene editing in living cells by inducing targeted DNA DSBs at specific genomic loci, and are composed of zinc finger proteins (ZFPs) Cys2-His2 and a non-specific DNA restriction enzyme (FokI endonuclease) [28]."

However, this is not correct. FokI is a specific restriction endonuclease. Only the nuclease domain of this enzyme is used to construct the ZFNs (and TALENs), which is indeed, non-specific without the ZF or DNA TALE recognition domains.

Response: Kind suggestions and accepted.

Revision explanations: As you suggested, we have revised the description of the ZFN technical principle, hoping to eliminate the misunderstanding caused by the problem of expression (Line 154-155).

  1. Figure 4. The authors state: "Sweden, Netherlands, and France are potential 276 competitors in gene editing research with low publication volume but high quality”

I wonder why these countries have been selected to mention. I would say, that based on the data in the table e.g. Germany, South Corea, and Italy seem to have larger contribution than Switzerland.

It is not only the citation number that characterizes the quality of a publication. However, it can be one of the measures. Nevertheless, the comparison of the average number of citations can be misleading when the number of publications taken into account is very different. The average number of citations is very sensitive to the deviations when low number of publications is included, but almost completely insensitive when large number of publications are included in the evaluation. I would suggest to include the ranges of the citations as a kind of error bars in the figure. It could be also interesting to select e.g. the 100 most cited publications from each country for comparison in another figure panel.

Response: Kind suggestions and accepted.

Revision explanations: As you suggested, we have re-evaluated the quality of each country's publications and revised the countries to Germany, South Korea and France (Line 281-283), and use the term “influence” to replace the “quality” which will be more objective. A separate analysis and comparison of the special case - Sweden was added in line 283-289.

  1. When analysing the authors’ contributions the authors state: "Doudna and Charpentier 320 shared the 2020 Nobel Prize in Chemistry for their discovery of the CRISPR/Cas9 genetic 321 scissors, but the latter doesn’t enter the top 20 researchers in terms of publication volume."

This is an important finding and shows that the publication volume of an author does not necessarily reflect the real contribution to the field. The collaboration of researchers – sometimes even from very different research fields – may lead to new results. However, the collaboration partner shall not be selected based on the publication volume – which might partially be suggested by the manuscript.

Response: Kind suggestions and accepted.

Revision explanations: As you suggested, we have discussed that there is no absolute correlation between outstanding contributors and their number of publications (Line 336-341).

  1. P20. The authors state: "According to Figure 9 (Supplementary Table 7), the keywords in regulatory research of GE are targeted mutagenesis, DNA, homologous recombination, protein, RNA, human cell, and so on, indicating that the focus of gene editing supervision is not only the technical details at the molecular level, but also ethical issues of GE technology applied to human embryos and germ cells."

Unfortunately, the keywords like off-target effects or efficiency or safety are missing or weak in Figure 9 and Table S7. In my opinion it needs to be discussed whether sometimes the potential profit from using e.g. gene edited crops is considered to be more important than the safety, which can’t be assessed in a short term experiment.

Response: Kind suggestions and accepted.

Revision explanations: As you suggested, we have discussed and analyzed the reasons for the missing and weak of keywords in regulation, and explained the specific reasons for the phenomenon of keywords missing and weak in related regulatory studies by taking gene-edited crops as an example (Line 409-420).

  1. The manuscript needs to be carefully checked for typos.

Response: As you suggested, we have carefully checked for typos.

This manuscript is a resubmission of an earlier submission. The following is a list of the peer review reports and author responses from that submission.

Round 1

Reviewer 1 Report

The paper by Wei and coworkers focuses on the analysis of the literature on gene editing, starting from the initial use of meganucleases and encompassing ZNF, Talen and CRISPR techniques. It clearly shows the rise of the CRISPR technology which has essentially terminated the use of the two other approaches, due to its higher simplicity and low costs. The increase in the number of papers parallels the continuous addition of new tools based on CRISPR technology, including the identification and adaptation of other Cas enzymes for gene editing and the implementation of Cas-based techniques such as Base Editors and Prime Editing. The paper mentions other aims of the technology, including gene regulation, mainly in the three fields of medicine, agriculture and industry. Literature analysis is put in correlation with the great advances made in the last 10 years since the discovery of Cas9, which was the first programmable Cas nuclease adapted to gene editing.

The paper is well written. The only suggestion I can do is to modify the Discussion section, which is too much dedicated to the CRISPR technology itself and not to the bibliographic analysis. This aspect has already been widely reported by many reviews available in the literature. I suggest that the Discussion should focus more on the bibliographic analysis, which is the real topic of the paper.

At lane 227, when speaking about DMD, I would add the word “partial”, as the approach used in the cited references does not allow a complete rescue of this pathology.

Author Response

1. The paper is well written. The only suggestion I can do is to modify the Discussion section, which is too much dedicated to the CRISPR technology itself and not to the bibliographic analysis. This aspect has already been widely reported by many reviews available in the literature. I suggest that the Discussion should focus more on the bibliographic analysis, which is the real topic of the paper. Response: Good suggestion and agreed. Revision explanations: As described by Small (1994), bibliometrics never claimed to offer insights into scientific knowledge, but the tools of bibliometrics can in fact be put at the service of advancing scientific knowledge in actual practice. With this in mind, we have rewrote the Discussion part, focusing more on the demonstration of Bibliometrics and our data using in evaluating various facets of a scientific topic. (Seen from Line 397-414, 436-438) Ref: Small, H. A SCI-map case study: building a map of AIDS research. Scientometrics. 1994, 30, 229 – 241. 2. At lane 227, when speaking about DMD, I would add the word “partial”, as the approach used in the cited references does not allow a complete rescue of this pathology. Response: Good suggestion and accepted. Revision explanations: Seen from Line 246.

Reviewer 2 Report

Review of Manuscript ID: cells-1661354

Title: The bibliometrics-based Investigation on Status of Gene Editing Technology Research in the Worldwide

The present manuscript reports the results of a bibliographical analysis regarding gene editing (GE), based on an in-depth search in the web of science. The authors identified a total of 13980 articles related to gene editing aspects. The authors analysed the dataset with regard to chronological development of GE, application areas, countries in which GE research had been published, research institutions, main research groups, „prolific authors“, collaborative networks, etc.

I have serious problems with the current paper.

1.    GE comprises ZFNs, TALEN and CRISPR with its various modifications. The current text is clearly biased for CRISPR, while ZFNs and TALEN are largely neglected or at least are not appropriately described. A more differentiated view on the three GE tools is needed.

2.    The chapter on the application areas of GE is very superficial and not really informative. For example, a table showing prominent examples of successful GE applications would be helpful for the readers.

3.    The chapter on GE research among countries (3.4) is particularly obscure. The authors operate with terms such as „quality of literature, e.g. average citations“ and name countries „technical leaders“ or technology activists“ (obviously only second category countries). I don`t think this type of categorization without real good definition should be allowed in a serious scientific journal.

4.   Similar holds true for chapter 3.5. Journals from Springer-Nature publish annual ranking lists for research institutions around the globe, also for certain research areas. Even these lists have their limitations, but the underlying scoring mechanisms are at least transparent. These statistics seem to be much more reliable than this one here. Overall, I found the discussion in this chapter particularly strange and and not very informative.

5.   The overarching discussion in chapter 4 is disappointing and lacks important aspects of GE, f.ex. potential legislation, differentiation from classical GMOs, pre- and clinical application, etc.

6.   The paper contains unclear sentences throughout the manuscript (f.ex. l. 183/184, 291-293,321-323,379-380, etc). Also the English language has weaknesses throughout the manuscript.

In conclusion, the current paper lacks to a large extent internationally accepted scientific standards.

Author Response

  1. GE comprises ZFNs, TALEN and CRISPR with its various modifications. The current text is clearly biased for CRISPR, while ZFNs and TALEN are largely neglected or at least are not appropriately described. A more differentiated view on the three GE tools is needed.

Response: Kind suggestions and accepted.

Revision explanations: As you suggested, we have made more differentiated views on Meganuclease, ZFNs, TALEN and CRISPR, and put them into the results section (Seen from Line 131-132, 144-148, 162-166, 184-186).

  1. The chapter on the application areas of GE is very superficial and not really informative. For example, a table showing prominent examples of successful GE applications would be helpful for the readers.

Response: Good suggestions and accepted.

Revision explanations: As you suggested, we have added some application cases of gene editing technology and inserted a table (Table 1) as the supplements to the text, in order to help the readers to retrieve the required information more clearly and conveniently.

  1. The chapter on GE research among countries (3.4) is particularly obscure. The authors operate with terms such as „quality of literature, e.g. average citations“ and name countries „technical leaders“ or technology activists“ (obviously only second category countries). I don`t think this type of categorization without real good definition should be allowed in a serious scientific journal.

Response: Thanks for the thoughts-provoking question.

Revision explanations: It is widely agreed that productivity and impact of the articles issued by actors in levels of countries, institutions and researchers are two aspects to evaluate the competitiveness of the actors (1-2). In the typical bibliometric analysis, the average citation is a robust index to quantify the impact and quality of scientific research (3-4). There are four categories of countries based on the volume and citations of the publications. We use more descriptive sentences instead of the terms "technical leaders" or "technology activists" according to the reviewer's suggestion. (Seen from Line 270-273, 275-277, 292-294)

Ref:

(1) Diniz, P., et al. A complementary index to quantify an individual's scientific research output. Scientometrics, 2005 , 68, 179-189.

(2) Ball, P. Index to aims for fair ranking of scientists. Nature, 2005, 436 ,7053, 900.

(3) Garfield, E. & Melino, G. The growth of the cell death field: an analysis from the ISI-Science Citation Index. Cell Death & Differentiation, 1997,4,352–361.

(4) Lawrence, P. A. The politics of publication. Nature, 2003,422, 259–261.

  1. 4. Similar holds true for chapter 3.5. Journals from Springer-Nature publish annual ranking lists for research institutions around the globe, also for certain research areas. Even these lists have their limitations, but the underlying scoring mechanisms are at least transparent. These statistics seem to be much more reliable than this one here. Overall, I found the discussion in this chapter particularly strange and not very informative.

Response: Thanks for the questions.

Revision explanations: The bibliometric analysis has very transparent scoring mechanisms and indexes on ranking the institutions and scientists, which can be seen clearly from Table 2 and Table 3. All the data were not collected manually, but from the Science Citation Index Expanded (SCIE) database in Web of Science, based on the gene- editing- related queries (Supplementary Table 1). Besides, the main findings about the ranking of institutes and authors in this paper agree with some previous studies to a large extent (1-3). We rewrite the Discussion part, focusing more on the demonstration of Bibliometrics and our data using in evaluating various facets of a scientific topic. (Seen from Line 397-414, 436-438)

Ref:

(1) Huang, Y., et al. Collaborative networks in gene editing. Nature biotechnology. 2019, 37, 1107-1109.

(2) Zhou, W., et al. A Decade of CRISPR Gene Editing in China and Beyond: A Scientometric Landscape. The CRISPR Journal, 2021,3, 313-320.

(3) Marshall, A., Trends in biotech literature 2008. Nature Biotechnology, 2009, 27,9, 789.

  1. The overarching discussion in chapter 4 is disappointing and lacks important aspects of GE, f.ex. potential legislation, differentiation from classical GMOs, pre- and clinical application, etc.

Response: Good suggestions and accepted.

Revision explanations: In the revised manuscript, we have added the discussion of GE in aspects of legislation and application, compared to GMOs. (Seen from line 467-480)

  1. The paper contains unclear sentences throughout the manuscript (f.ex. l. 183/184, 291-293,321-323,379-380, etc). Also the English language has weaknesses throughout the manuscript.

Response: Good suggestions and agreed. All the lines mentioned by the reviewer have been revised, and the whole manuscript has been polished by the authors and our foreign cooperator.

Reviewer 3 Report

1. Can the authors comment on this point that how their analysis provides insights into the future trajectory of development and application of the technology in various fields and how from their study this will be helpful to popularization of gene editing technology.

2. Check line 66 (CRISPR is written in different font)

3. The authors should diagrammatically explain the Gene Editing Technologies so that a reader can understand it well.

4. Under the section “Main Application Directions of Gene Editing Technology” the authors should further explain by dividing industry and medicine to further subsections (Examples: studies on different diseases using mice and Human or cells and their outcome and future directions)

Author Response

  1. Can the authors comment on this point that how their analysis provides insights into the future trajectory of development and application of the technology in various fields and how from their study this will be helpful to popularization of gene editing technology.

Response: Good suggestion and agreed.

Revision explanations:

    Recognizing that the purpose of publishing technical papers is to communicate novel findings and innovations, each peer-reviewed research report may be considered to embody a quantum of new information representing a scientific or technical advance. Integrated over large numbers of papers, representing a distribution of large and small advances, the total number of publications on a particular technology provides a metric for the advance of that technology.

    To be specific, the latest keywords in the timeline and achievements of the leading authors may chart a course for the future of a certain area. In this sense, the four aspects of future strategies to promote the development of gene editing technology are proposed in this study. Besides, the bibliometric analysis can illustrate the performance of key actors, the trend of research hotspots and so on, which will help the readers get a full view of the status on gene editing technology and its application in a short time.

  1. Check line 66 (CRISPR is written in different font)

Response: Good suggestion and agreed. (Seen from Line 65)

  1. The authors should diagrammatically explain the Gene Editing Technologies so that a reader can understand it well.

Response: Good suggestion and agreed.

Revision explanations: To better understand the gene editing technologies from the study, a schematic diagram of four types of gene editing technologies was added in the revised manuscript ( Seen from Page 5, Figure 3).

  1. Under the section “Main Application Directions of Gene Editing Technology” the authors should further explain by dividing industry and medicine to further subsections (Examples: studies on different diseases using mice and Human or cells and their outcome and future directions)

Response: Good suggestion and agreed.

Revision explanations: As you suggested, we have added some application cases of gene editing technology, especially in the medicine and health subsections, and inserted a table (Table 1) as the supplements to the text.

Round 2

Reviewer 2 Report

The paper has undergone revision by the authors, some of the main weaknesses has been ameliorated, but there is no substantial improvement of the paper. There are still major problems with some important issues, particularly  the description of the GE application areas, which is fragmentary and non-representative, and the term quality of research outcome. Therefore I still hold up my critique from the first version.